# Latent Distance Estimation for Random Geometric Graphs

**Ernesto Araya**
Laboratoire de Mathématiques d'Orsay (LMO)
Université Paris-Sud
91405 Orsay Cedex, France
ernesto.araya-valdivia@u-psud.fr

**Yohann De Castro**
Institut Camille Jordan
École Centrale de Lyon
69134 Écully, France
yohann.de-castro@ec-lyon.fr

## Abstract

Random geometric graphs are a popular choice for a latent points generative model for networks. Their definition is based on a sample of $n$ points $X_1, X_2, \cdots, X_n$ on the Euclidean sphere $\mathbb{S}^{d-1}$ which represents the latent positions of nodes of the network. The connection probabilities between the nodes are determined by an unknown function (referred to as the "link" function) evaluated at the distance between the latent points. We introduce a spectral estimator of the pairwise distance between latent points and we prove that its rate of convergence is the same as the nonparametric estimation of a function on $\mathbb{S}^{d-1}$, up to a logarithmic factor. In addition, we provide an efficient spectral algorithm to compute this estimator without any knowledge on the nonparametric link function. As a byproduct, our method can also consistently estimate the dimension $d$ of the latent space.

## 1 Introduction

Random geometric graph (RGG) models have received attention lately as alternative to some simpler yet unrealistic models as the ubiquitous Erdös-Rényi model [12]. They are generative latent point models for graphs, where it is assumed that each node has associated a latent point in a metric space (usually the Euclidean unit sphere or the unit cube in $\mathbb{R}^d$) and the connection probability between two nodes depends on the position of their associated latent points. In many cases, the connection probability depends only on the distance between the latent points and it is determined by a one-dimensional "link" function.

Because of its geometric structure, this model is appealing for applications in wireless networks modeling [18], social networks [17] and biological networks [15], to name a few. In many of these real-world networks, the probability that a tie exists between two agents (nodes) depends on the similarity of their profiles. In other words, the connection probability depends on some notion of distance between the position of the agents in a metric space, which in the social network literature has been called the *social space*.

In the classical RGG model, as introduced by Gilbert in [13], we consider $n$ independent and identically distributed latent points $\{X_i\}_{i=1}^n$ in $\mathbb{R}^d$ and the construct the graph with vertex set $V = \{1, 2, \cdots, n\}$, where the node $i$ and $j$ are connected if and only if the Euclidean distance $\|X_i - X_j\|_d$ is smaller than a certain predefined threshold $\tau$. The seminal reference on the classical RGG model, from the probabilistic point of view, is the monograph [27]. Another good reference is the survey paper [30]. In such a case, the "link" function, which we have not yet formally defined, is the *threshold* function $\mathbb{1}_{t \leq \tau}(t)$. Otherwise stated, two points are connected only if their distance is smaller than $\tau$. In that case, all the randomness lies in the fact that we are sampling the latent points with a certain distribution. We choose to maintain the name of random geometric graphs for more general "link" functions.

The angular version of the RGG model has also received attention. On that model, the latent points are uniformly distributed on $\mathbb{S}^{d-1}$ (the unit sphere on $\mathbb{R}^d$), and two points are connected if their angle is bellow a certain threshold. This model has been used in the context of sensor and wireless networks [14]. In [9] the authors show that in when the size of the graph $n$ is fixed and the dimension $d$ goes to infinity, the RGG model on the sphere is indistinguishable from the Erdös-Renyi model, in the sense that the total variation distance between both graph distributions converges to zero. On the other hand, in [5] the authors prove that in the dense case if $d$ satisfy a bound with respect to $n$ (specifically, if $d/n^3 \to 0$) then we can distinguish between both models, by a counting the number of triangles. The angular RGG model has also been used in the context of approximate graph coloring [19].

We are interested in the problem of recovering the pairwise distances between the latent points $\{X_i\}_{i=1}^n$ for geometric graphs on $\mathbb{S}^{d-1}$ given a single observation of the network. We limit ourselves to the case when the network is a simple graph. Furthermore, we will assume that the dimension $d$ is fixed and that the "link" function is not known. This problem and some of its variants have been studied for different versions of the model and under a different set of hypothesis, see for example the recent work [1] and the references therein. In that work the authors propose a method for estimating the latent distances based on the graph theoretic distance between two nodes (that is the length of the shortest path between the nodes). Independently, in [10] the authors develop a similar method which has slightly less recovery error, but for a less general model. In both cases, the authors consider the cube in $\mathbb{R}^d$ (or the whole $\mathbb{R}^d$) but not the sphere. Our strategy is similar to the one developed in [28], where they considered the latent point estimation problem in *random dot product graphs*, which is a more restricted model compared to the one considered here. However, they considered more general Euclidean spaces and latent points distributions other than the uniform. Similar ideas has been used in the context of vertex classification for latent position graphs [29].

We will use the notion of graphon function to formalize the concept of "link" function. Graphons are central objects to the theory of dense graph limits. They were introduced by Lovász and Szegedy in [25] and further developed in a series of papers, see [3],[4]. Formally, they are symmetric kernels that take values in $[0, 1]$, thus they will act as the "link" function for the latent points. The spectrum of the graphon is defined as the spectrum of an associated integral operator, as in [24, Chap.7]. In this paper, they will play the role of limit models for the adjacency matrix of a graph, when the size goes to infinity. This is justified in light of the work of Koltchinskii and Giné [22] and Koltchinskii [21]. In particular, the adjacency matrix of the observed graph can be though as a finite perturbed version of this operator, combining results from [22] and [2].

We will focus on the case of dense graphs on the sphere $\mathbb{S}^{d-1}$ where the connection probability depends only on the angle between two nodes. This allows us to use the harmonic analysis on the sphere to have a nice characterization of the graphon spectrum, which has a very particular structure. More specifically, the following two key elements holds: first of all, the basis of eigenfunctions is fixed (do not depend on the particular graphon considered) and equal to the well-known spherical harmonic polynomials. Secondly, the multiplicity of each eigenvalue is determined by a sequence of integers that depends only on the dimension $d$ of the sphere and is given by a known formula and the associated eigenspaces are composed by spherical harmonics of the same polynomial degree.

The graphon eigenspace composed only with linear eigenfunctions (harmonic polynomials of degree one) will play an important role in the latent distances matrix recovery as all the information we need to reconstruct the distances matrix is contained in those eigenfunctions. We will prove that it is possible to approximately recover this information from the observed adjacency matrix of the graph under regularity conditions (of the Sobolev type) on the graphon and assuming an eigenvalue gap condition (similar hypotheses are made in [6] in the context of matrix estimation and in [23] in the context of manifold learning). We do this by proving that a suitable projection of the adjacency matrix, onto a space generated by exactly $d$ of its eigenvectors, approximates well the latent distances matrix considering the mean squared error in the Frobenius norm. We give nonasymptotic bounds for this quantity obtaining the same rate as the nonparametric rate of estimation of a function on the sphere $\mathbb{S}^{d-1}$, see [11, Chp.2] for example. Our approach includes the adaptation of some perturbation theorems for matrix projections from the orthogonal to a "nearly" orthogonal case, which combined with concentration inequalities for the spectrum gives a probabilistic finite sample bound, which is novel to the best of our knowledge. More specifically, we prove concentration inequalities for the sampled eigenfunctions of the integral operator associated to a geometric graphon, which are not necessarily orthogonal as vectors in $\mathbb{R}^n$. Our method shares some similarities with the celebrated

USVT method, introduced by Chatterjee in [6], but in that case they obtained an estimator of the probability matrix described in Section 2.2 and not of the population Gram matrix as our method. We develop an efficient algorithm, which we call Harmonic EigenCluster(HEiC) to reconstruct the latent positions from the data and illustrate its usefulness with synthetic data.

## 2 Preliminaries

### 2.1 Notation

We will consider $\mathbb{R}^d$ with the Euclidean norm $\|\cdot\|$ and the Euclidean scalar product $\langle\,,\,\rangle$. We define the sphere $\mathbb{S}^{d-1} := \{x \in \mathbb{R}^d : \|x\| = 1\}$. For a set $A \subset \mathbb{R}$ its diameter $diam(A) := \sup_{x,y \in A} |x - y|$ and if $B \subset \mathbb{R}$ the distance between $A$ and $B$ is $dist(A, B) := \inf_{x \in A, y \in B} |x - y|$. We will use $\|\cdot\|_F$ the Frobenius norm for matrices and $\|\cdot\|_{op}$ for the operator norm. The identity matrix in $\mathbb{R}^{d \times d}$ will be $\mathrm{Id}_d$. If $X$ is a real valued random variable and $\alpha \in (0, 1)$, $X \leq_\alpha C$ means that $\mathbb{P}(X \leq C) \geq 1 - \alpha$.

### 2.2 Generative model

We describe the generative model for networks which is a generalization of the classical random geometric graph model introduced by Gilbert in [13]. We base our definition on the $W$-random graph model described in [24, Sec. 10.1]. The central objects will be graphon functions on the sphere, which are symmetric measurable functions of the form $W : \mathbb{S}^{d-1} \times \mathbb{S}^{d-1} \to [0, 1]$. Throughout this paper, we consider the measurable space $(\mathbb{S}^{d-1}, \sigma)$, where $\sigma$ is the uniform measure on the sphere. On $\mathbb{S}^{d-1} \times \mathbb{S}^{d-1}$ we consider the product measure $\sigma \times \sigma$.

To generate a simple graph from a graphon function, we first sample $n$ points $\{X_i\}_{i=1}^n$ independently on the sphere $\mathbb{S}^{d-1}$, according to the uniform measure $\sigma$. These are the so-called latent points. Secondly, we construct the matrix of distances between these points, called the *Gram matrix* $\mathcal{G}^*$ (we will often call it population Gram matrix) defined by

$$\mathcal{G}^*_{ij} := \langle X_i, X_j \rangle$$

and the so-called *probability matrix*

$$\Theta_{ij} = \rho_n W(X_i, X_j)$$

which is also a $n \times n$ matrix. The function $W$ gives the precise meaning for the "link" function, because it determines the connection probability between $X_i$ and $X_j$. The introduction of the scale parameter $0 < \rho_n \leq 1$ allow us to control the edge density of the sampled graph given a function $W$, see [20] for instance. The case $\rho_n = 1$ corresponds to the dense case (the parameter $\Theta_{ij}$ do not depend on $n$) and when $\rho_n \to 0$ the graph will be sparser. Our main results will hold in the regime $\rho_n = \Omega(\frac{\log n}{n})$, which we call *relatively sparse*. Most of the time we will work with the normalized version of the probability matrix $T_n := \frac{1}{n}\Theta$. If there exists a function $f : [-1, 1] \to [0, 1]$ such that $W(x, y) = f(\langle x, y \rangle)$ for all $x, y \in \mathbb{S}^{d-1}$ we will say that $W$ is a geometric graphon.

Finally, we define the random adjacency matrix $\hat{T}_n$, which is a $n \times n$ symmetric random matrix that has independent entries (except for the symmetry constraint $\hat{T}_n = \hat{T}_n^T$), conditional on the probability matrix, with laws

$$n(\hat{T}_n)_{ij} \sim \mathcal{B}(\Theta_{ij})$$

where $\mathcal{B}(m)$ is the Bernoulli distribution with mean parameter $m$. Since the probability matrix contains the mean parameters for the Bernoulli distributions that define the random *adjacency* matrix it has been also called the *parameter matrix* [6]. Observe that the classical RGG model on the sphere is a particular case of the described $W$-random graph model when $W(x, y) = \mathbb{1}_{\langle x,y \rangle \leq \tau}$. In that case, since the entries of the probability matrix only have values in $\{0, 1\}$, the adjacency matrix and the probability matrix are equal. Depending on the context, we use $\hat{T}_n$ for the random matrix as described above or for an instance of this random matrix, that is for the adjacency matrix of the observed graph. This will be clear from the context.

It is worth noting that graphons can be, without loss of generality, defined in $[0, 1]^2$. The previous affirmation means that for any graphon there exists a graphon in $[0, 1]^2$ that generates the same distribution on graphs for any given number of nodes. However, in many cases the $[0, 1]^2$ representation

can be less revealing than other representations using a different underlying space. This is illustrated in the case of the *prefix attachment* model in [24, example 11.41].

In the sequel we use the notation $\lambda_0, \lambda_1, \cdots, \lambda_{n-1}$ for the eigenvalues of the normalized probability matrix $T_n$. Similarly, we denote by $\hat{\lambda}_0, \hat{\lambda}_1, \cdots, \hat{\lambda}_{n-1}$ the eigenvalues of the matrix $\hat{T}_n$. We recall that $T_n$ (resp. $\hat{T}_n$) and $\frac{1}{\rho_n}T_n$ (resp. $\frac{1}{\rho_n}\hat{T}_n$) have the same set of eigenvectors. We will denote by $v_j$ for $1 \leq j \leq n$ the eigenvector of $T_n$ associated to $\lambda_j$, which is also the eigenvector of $\frac{1}{\rho_n}T_n$ associated to $\frac{1}{\rho_n}\lambda_j$. Similarly, we denote by $\hat{v}_j$ to the eigenvector associated to the eigenvalue $\rho_n\hat{\lambda}_j$ of $\hat{T}_n$.

Our main result is that we can recover the Gram matrix using the eigenvectors of $\hat{T}_n$ as follows

**Theorem 1** (Informal statement). *There exists a constant $c_1 > 0$ that depends only on the dimension $d$ such that the following is true. Given a graphon $W$ on the sphere such that $W(x,y) = f(\langle x, y \rangle)$ with $f : [-1, 1] \to [0, 1]$ unknown, which satisfies an eigenvalue gap condition and has Sobolev regularity $s$, there exists a subset of the eigenvectors of $\hat{T}_n$, such that $\hat{\mathcal{G}} := \frac{1}{c_1}\hat{V}\hat{V}^T$ converges to the population Gram matrix $\mathcal{G}^* := \frac{1}{n}(\langle X_i, X_j \rangle)_{i,j}$ at rate $n^{\frac{-s}{2s+d-1}}$ (up to a log factor). This estimate $\hat{V}\hat{V}^T$ can be found in linear time given the spectral decomposition of $\hat{T}_n$.*

We will say that a geometric graphon $W(x,y) = f(\langle x, y \rangle)$ on $\mathbb{S}^{d-1}$ has regularity $s$ if $f$ belongs the weighted Sobolev space $Z_\gamma^s([-1, 1])$ with weight function $w_\gamma(t) = (1 - t)^{\gamma - \frac{1}{2}}$, as defined in [26]. In order to make the statement of 1 rigorous, we need to precise the eigenvalue gap condition and define the graphon eigensystem.

## 2.3 Geometric graphon eigensystem

Here we gather some asymptotic and concentration properties for the eigenvalues and eigenfunctions of the matrices $\hat{T}_n, T_n$ and the operator $T_W$, which allows us to recover the Gram matrix from data. The key fact is that the eigenvalues (resp. eigenvectors) of the matrix $\frac{1}{\rho_n}\hat{T}_n$ and $\frac{1}{\rho_n}T_n$ converge to the eigenvalues (resp. sampled eigenfunctions) of the integral operator $T_W : L^2(\mathbb{S}^{d-1}) \to L^2(\mathbb{S}^{d-1})$

$$T_W g(x) = \int_{\mathbb{S}^{d-1}} g(y) W(x, y) d\sigma(y)$$

which is compact [16, Sec.6, example 1] and self-adjoint (which follows directly from the symmetry of $W$). Then by a classic theorem in functional analysis [16, Sec.6, Thm. 1.8] its spectrum is a discrete set $\{\lambda_k^*\}_{k \in \mathbb{N}} \subset \mathbb{R}$ and its only accumulation point is zero. In consequence, we can see the spectra of $\hat{T}_n, T_n$ and $T_W$ (which we denote $\lambda(\hat{T}_n), \lambda(T_n)$ and $\lambda(T_W)$ resp.) as elements of the space $\mathcal{C}_0$ of infinite sequences that converge to 0 (where we complete the finite sequences with zeros). It is worth noting that in the case of geometric graphons with regularity $s$ (in the Sobolev sense defined above) the rate of convergence of $\lambda(T_W)$ is determined by the regularity parameter $s$. We have the following:

- The spectrum of $\lambda(\frac{1}{\rho_n}T_n)$ converges to $\lambda(T_W)$ (almost surely) in the $\delta_2$ metric, defined as follows

$$\delta_2(x, y) = \inf_{p \in \mathcal{P}} \sqrt{\sum_{i \in \mathbb{N}} (x_i - y_{p(i)})^2}$$

where $\mathcal{P}$ is the set of all permutations of the non-negative integers. This is proved in [22]. In [8] they prove the following

$$\delta_2\left(\lambda(\frac{1}{\rho_n}T_n), \lambda(T_W)\right) \leq_{\alpha/4} C\left(\frac{\log n}{n}\right)^{\frac{s}{2s+d-1}} \tag{1}$$

- Matrices $\hat{T}_n$ approach to matrix $T_n$ in operator norm as $n$ gets larger. Applying [2, Cor.3.3] to the centered matrix $Y = \hat{T}_n - T_n$ we get

$$\mathbb{E}(\|\hat{T}_n - T_n\|_{op}) \lesssim \frac{D_0}{n} + \frac{D_0^* \sqrt{\log n}}{n} \tag{2}$$

where $\lesssim$ denotes inequality up to constant factors, $D_0 = \max_{0 \le i \le n} \sum_{j=1}^{n} Y_{ij}(1 - Y_{ij})$ and $D_0^* = \max_{ij}|Y_{ij}|$. We clearly have that $D_0 = \mathcal{O}(n\rho_n)$ and $D_0^* \le 1$, which implies that

$$\mathbb{E}\|\hat{T}_n - T_n\|_{op} \lesssim \max\left\{\frac{\rho_n}{\sqrt{n}}, \frac{\sqrt{\log n}}{n}\right\}$$

We see that this inequality do not improve if $\rho_n$ is smaller than in the relatively sparse case, that is $\rho_n = \Omega(\frac{\log n}{n})$. We prove that, as a corollary of the results in [2], we have

$$\frac{1}{\rho_n}\|\hat{T}_n - T_n\|_{op} \le_{\alpha/4} C \max\left\{\frac{1}{\sqrt{\rho_n n}}, \frac{\sqrt{\log n}}{\rho_n n}\right\} \tag{3}$$

An analogous bound can be obtained for the Frobenius norm replacing $\hat{T}_n$ with $\hat{T}_n^{\text{usvt}}$ the USVT estimator defined in [6]. For our main results, Proposition 3 and Theorem 4 the operator norm bound will suffice.

A remarkable fact in the case of geometric graphons on $\mathbb{S}^{d-1}$ is that the eigenfunctions $\{\phi_k\}_{k \in \mathbb{N}}$ of the integral operator $T_W$ are a fixed set that do not depend on the particular function $f$ considered. This comes from the fact that $T_W$ is a convolution operator on the sphere and its eigenfunctions are the well-known *spherical harmonics* of dimension $d$, which are harmonic polynomials in $d$ variables defined on $\mathbb{S}^{d-1}$ corresponding to the eigenfunctions of the Laplace-Beltrami operator on the sphere. This follows from [7, Thm.1.4.5] and from the Funck-Hecke formula given in [7, Thm.1.2.9]. Let $d_k$ denote the dimension of the $k$-th spherical harmonic space. It is well-known [7, Cor.1.1.4] that $d_0 = 1$, $d_1 = d$ and $d_k = \binom{k+d-1}{k} - \binom{k+d-3}{k-2}$. Another important fact, known as the *addition theorem* [7, Lem.1.2.3 and Thm.1.2.6], is that

$$\sum_{i=d_{k-1}}^{d_k} \phi_j(x)\phi_j(y) = c_k G_k^\gamma(\langle x, y \rangle)$$

where $G_k^\gamma$ are the Gegenbauer polynomials of degree $k$ with parameter $\gamma = \frac{d-2}{2}$ and $c_k = \frac{2k+d-2}{d-2}$.

The Gegenbauer polynomial of degree one is $G_1^\gamma(t) = 2\gamma t$ (see [7, Appendix B2]), hence we have $G_1^\gamma(\langle X_i, X_j \rangle) = 2\gamma\langle X_i, X_j \rangle$ for every $i$ and $j$. In consequence, by the addition theorem

$$G_1^\gamma(\langle X_i, X_j \rangle) = \frac{1}{c_1}\sum_{k=1}^{d}\phi_k(X_i)\phi_k(X_j)$$

where we recall that $d_1 = d$. This implies the following relation for the Gram matrix, observing that $2\gamma c_1 = d$

$$\mathcal{G}^* := \frac{1}{n}(\langle X_i, X_j \rangle)_{i,j} = \frac{1}{2\gamma c_1}\sum_{j=1}^{d}v_j^* v_j^{*T} = \frac{1}{d}V^* V^{*T} \tag{4}$$

where $v_j^*$ is the $\mathbb{R}^n$ vector with $i$-th coordinate $\phi_j(X_i)/\sqrt{n}$ and $V^*$ is the matrix with columns $v_j^*$. In a similar way, we define for any matrix $U$ in $\mathbb{R}^{n \times d}$ with columns $u_1, u_2, \cdots, u_d$, the matrix $\mathcal{G}_U := \frac{1}{d}UU^T$. As part of our main theorem we prove that for $n$ large enough there exists a matrix $\hat{V}$ in $\mathbb{R}^{n \times d}$ where each column is one of the eigenvector of $\hat{T}_n$, such that $\hat{\mathcal{G}} := \mathcal{G}_{\hat{V}}$ approximates $\mathcal{G}^*$ well, in the sense that the norm $\|\hat{\mathcal{G}} - \mathcal{G}^*\|_F$ converges to 0 at a rate which is that of the nonparametric estimation of a function on $\mathbb{S}^{d-1}$.

## 2.4 Eigenvalue gap condition

In this section we describe one of our main hypotheses on $W$ needed to ensure that the space $\text{span}\{v_1^*, v_2^*, \cdots, v_d^*\}$ can be effectively recovered with the vectors $\hat{v}_1, \hat{v}_2, \cdots, \hat{v}_d$ using our algorithm. Informally, we assume that the eigenvalue $\lambda_1^*$ is sufficiently isolated from the rest of the spectrum of $T_W$ (not counting multiplicity). We assume without loss of generality that $\lambda_1^* = \lambda_2^* = \cdots = \lambda_{d_1}^*$. Given a geometric graphon $W$, we define the *spectral gap* of $W$ relative to the eigenvalue $\lambda_1^*$ by

$$\text{Gap}_1(W) := \min_{j \notin \{1, \cdots, d_1\}} |\lambda_1^* - \lambda_j^*|$$

which quantifies the distance between the eigenvalue $\lambda_1^*$ and the rest of the spectrum. In particular, we have the following elementary proposition.

**Proposition 2.** *It holds that* $\mathrm{Gap}_1(W) = 0$ *if and only if there exists* $j \notin \{1, \cdots, d_1\}$ *such that* $\lambda_j^* = \lambda_1^*$ *or* $\lambda_1^* = 0$.

*Proof.* Observe that the unique accumulation point of the spectrum of $T_W$ is zero. The proposition follows from this observation. $\square$

To recover the population Gram matrix $\mathcal{G}^*$ with our Gram matrix estimator $\hat{\mathcal{G}}$ we require the spectral gap $\Delta^* := \mathrm{Gap}_1(W)$ to be different from 0. This assumption have been made before in the literature, in results that are based in some version of the Davis-Kahan $\sin\theta$ theorem (see for instance [6], [23], [29]). More precisely, our results will hold on the following event

$$\mathcal{E} := \left\{ \delta_2\Big(\lambda\big(\frac{1}{\rho_n}T_n\big), \lambda(T_W)\Big) \vee \frac{2^{\frac{9}{2}}\sqrt{d}}{\rho_n\Delta^*}\|T_n - \hat{T}_n\|_{op} \leq \frac{\Delta^*}{4} \right\},$$

for which we prove the following: given an arbitrary $\alpha$ we have that

$$\mathbb{P}(\mathcal{E}) \geq 1 - \frac{\alpha}{2}$$

for $n$ large enough (depending on $W$ and $\alpha$). This dependence can be made explicit using (1) and (3)

$$\max\left\{\sqrt{\frac{\rho_n}{n}}, \frac{\sqrt{\log n}}{n}\right\} \leq \frac{\Delta^{*2}}{2^{15/2}C\sqrt{d}} \quad \text{and} \quad \frac{\log n}{n} \leq \Big(\frac{\Delta^*}{8C'}\Big)^{\frac{2s+d-1}{s}}$$

where $C, C' > 0$. The following theorems are the main results of this paper. Their proofs can be found in the supplementary material.

**Proposition 3.** *On the event* $\mathcal{E}$, *there exists one and only one set* $\Lambda_1$, *consisting of* $d$ *eigenvalues of* $\hat{T}_n$, *whose diameter is smaller than* $\rho_n\Delta^*/2$ *and whose distance to the rest of the spectrum of* $\hat{T}_n$ *is at least* $\rho_n\Delta^*/2$. *Furthermore, on the event* $\mathcal{E}$, *our algorithm (Algorithm 1) returns the matrix* $\hat{\mathcal{G}} = (1/c_1)\hat{V}\hat{V}^T$, *where* $\hat{V}$ *has by columns the eigenvectors corresponding to the eigenvalues on* $\Lambda_1$.

**Theorem 4.** *Let* $W$ *be a regular geometric graphon on* $\mathbb{S}^{d-1}$ *with regularity parameter* $s$ *and such that* $\Delta^* > 0$. *Then there exists a set of eigenvectors* $\hat{v}_1, \cdots, \hat{v}_d$ *of* $\hat{T}_n$ *such that*

$$\|\mathcal{G}^* - \hat{\mathcal{G}}\|_F = O(n^{-\frac{s}{2s+d-1}})$$

*where* $\hat{\mathcal{G}} = \mathcal{G}_{\hat{V}}$ *and* $\hat{V}$ *is the matrix with columns* $\hat{v}_1, \cdots, \hat{v}_d$. *Moreover, this rate is the minimax rate of nonparametric estimation of a regression function* $f$ *with Sobolev regularity* $s$ *in dimension* $d - 1$.

The condition $\Delta^* > 0$ allow us to use Davis-Kahan type results for matrix perturbation to prove Theorem 4. With this and concentration for the spectrum we are able to control with high probability the terms $\|\hat{\mathcal{G}} - \mathcal{G}\|_F$ and $\|\mathcal{G} - \mathcal{G}^*\|_F$. The rate of nonparametric estimation of a function in $S^{d-1}$ can be found in [11, Chp.2].

## 3 Algorithms

The Harmonic EigenCluster algorithm(HEiC) (see Algorithm 1 below) receives the observed adjacency matrix $\hat{T}_n$ and the sphere dimension as its inputs to reconstruct the eigenspace associated to the eigenvalue $\lambda_1^*$. In order to do so, the algorithm selects $d$ vectors in the set $\hat{v}_1, \hat{v}_2, \cdots \hat{v}_n$, whose linear span is close to the span of the vectors $v_1^*, v_2^*, \cdots, v_d^*$ defined in Section 2.3. The main idea is to find a subset of $\{\hat{\lambda}_0, \hat{\lambda}_2, \cdots, \hat{\lambda}_{n-1}\}$, which we call $\Lambda_1$, consisting on $d_1$ elements (recall that $d_1 = d$) and where all its elements are close to $\lambda_1^*$. This can be done assuming that the event $\mathcal{E}$ defined above holds (which occurs with high probability). Once we have the set $\Lambda_1$, we return the span of the eigenvectors associated to the eigenvalues in $\Lambda_1$.

For a given set of indices $i_1, \cdots, i_d$ we define

$$\mathrm{Gap}_1(\hat{T}_n; i_1, \cdots, i_d) := \min_{i \notin \{i_1, \cdots, i_d\}} \max_{j \in \{i_1, \cdots, i_j\}} |\hat{\lambda}_j - \hat{\lambda}_i|$$

and

$$\mathrm{Gap}_1(\hat{T}_n) := \max_{\{i_1, \cdots, i_d\} \in \mathcal{S}_d^n} \mathrm{Gap}_1(\hat{T}_n; i_1, \cdots, i_d)$$

---

**Algorithm 1:** Harmonic EigenCluster(HEiC) algorithm

---

**Input:** $(\hat{T}_n, d)$ adjacency matrix and sphere dimension

  $\Lambda^{\text{sort}} = \{\hat{\lambda}_1^{\text{sort}}, \cdots, \hat{\lambda}_{n-1}^{\text{sort}}\} \leftarrow$ eigenvalues of $\hat{T}_n$ sorted in decreasing order
  $\Lambda_1 \leftarrow \{\Lambda_1^{\text{sort}}, \cdots, \Lambda_{1+d}^{\text{sort}}\}$: where $\Lambda_i^{\text{sort}}$ is the $i$-th element in $\Lambda^{\text{sort}}$
  Initialize $i = 2$, gap $= \text{Gap}_1(\hat{T}_n; 1, 2, \cdots, d)$

**while** $i \leq n - d$ **do**
  **if** $\text{Gap}_1(\hat{T}_n; i, i + 1, \cdots, i + d) >$ gap **then**
    $\Lambda_1 \leftarrow \{\Lambda_i^{\text{sort}}, \cdots, \Lambda_{i+d}^{\text{sort}}\}$
  **end if**
  $i \leftarrow i + 1$
**end while**

**Return:** $\Lambda_1$, gap

---

where $\mathcal{S}_d^n$ contains all the subsets of $\{1, \cdots, n-1\}$ of size $d$. This definition parallels that of $\text{Gap}_1(W)$ for the graphon. Observe any set of indices in $\mathcal{S}_d^n$ will not include 0. Otherwise stated, we can leave $\hat{\lambda}_0^{\text{sort}}$ out of this definition and it will not be candidate to be in $\Lambda_1$. In the supplementary material we prove that the largest eigenvalue of the adjacency matrix will be close to the eigenvalue $\lambda_0^*$ and in consequence can not be close enough to $\lambda_1^*$ to be in the set $\Lambda_1$, given the definition of the event $\mathcal{E}$.

To compute $\text{Gap}_1(\hat{T}_n)$ we consider the set of eigenvalues $\hat{\lambda}_j$ ordered in decreasing order. We use the notation $\hat{\lambda}_j^{\text{sort}}$ to emphasize this fact. We define the right and left differences on the sorted set by

$$\text{left}(i) = |\hat{\lambda}_i^{\text{sort}} - \hat{\lambda}_{i-1}^{\text{sort}}|$$
$$\text{right}(i) = \text{left}(i+1)$$

where $\text{left}(\cdot)$ is defined for $1 \leq i \leq n$ and $\text{right}(\cdot)$ is defined for $0 \leq i \leq n-1$. With these definition, we have the following lemma, which we prove in the supplementary material.

**Lemma 5.** *On the event $\mathcal{E}$, the following equality holds*

$$\text{Gap}_1(\hat{T}_n) = \max \left\{ \max_{1 \leq i \leq n-d-1} \min \{\text{left}(i), \text{right}(i+d)\}, \text{left}(n-d+1) \right\}$$

The set $\Lambda_1$ has the form $\Lambda_1 = \{\hat{\lambda}_{i^*}^{\text{sort}}, \hat{\lambda}_{i^*+1}^{\text{sort}}, \cdots, \hat{\lambda}_{i^*+d}^{\text{sort}}\}$ for some $1 \leq i^* \leq n - d - 1$. We have that either

$$i^* = \underset{1 \leq i \leq n-d-1}{\arg\max} \min \{\text{left}(i), \text{right}(i+d)\}$$

or $i^* = n - d$ depending whether or not one has $\max_{1 \leq i \leq n-d-1} \min \{\text{left}(i), \text{right}(i+d)\} > \text{left}(n-d+1)$. The algorithm then constructs the matrix $\hat{V}$ having columns $\{\hat{v}_{i^*}, \hat{v}_{i^*+1}, \cdots, \hat{v}_{i^*+d}\}$ and returns $\hat{V}\hat{V}^T$.

It is worth noting that Algorithm 1 time complexity $n^3 + n$, where $n^3$ comes from the fact that we compute the eigenvalues and eigenvectors of the $n \times n$ matrix $\hat{T}_n$ and the linear term is because we explore the whole set of eigenvalues to find the maximum gap for the size $d$. In terms of space complexity the algorithm is $n^2$ because we need to store the matrix $\hat{T}_n$.

**Remark 1.** *If we change $\hat{T}_n$ in the input of Algorithm 1 to $\hat{T}_n^{\text{usvt}}$ (obtained by the USVT algorithm [6]) we predict that the algorithm will give similar results. This is because discarding some eigenvalues bellow a prescribed threshold do not have effect on our method. However, as preprocessing step the USVT might help in speeding up the eigenspace detection, but this step is already linear in time.*

### 3.1 Estimation of the dimension $d$

So far we have focused on the estimation of the population Gram matrix $\mathcal{G}^*$. We now give an algorithm to find the dimension $d$, when it is not provided as input. This method receives the

matrix $\hat{T}_n$ as input and uses Algorithm 1 as a subroutine to compute a score, which is simply the value of the variable $\mathrm{Gap}_1(\hat{T}_n)$ returned by Algorithm 1. We do this for each $d$ in a set of candidates, which we call $\mathcal{D}$. This set of candidates will be usually, but not necessarily, fixed to $\{1, 2, 3, \cdots, d_{max}\}$. Once we have computed the scores, we pick the candidate that have the maximum score.

Given the guarantees provided by Theorem 4, the previously described procedure will find the correct dimension, with high probability (on the event $\mathcal{E}$), if the true dimension of the graphon is on the candidate set $\mathcal{D}$. This will happen, in particular, if the assumptions of Theorem 4 are satisfied. We recall that the main hypothesis on the graphon is that the spectral gap $\mathrm{Gap}_1(W)$ should be different from 0.

## 4 Experiments

We generate synthetic data using different geometric graphons. In the first set of examples, we focus in recovering the Gram matrix when the dimension is provided. In the second set we tried to recover the dimension as well.

### 4.1 Recovering the Gram matrix

We start by considering the graphon $W_1(x, y) = \mathbb{1}_{\langle x,y \rangle \leq 0}$ which defines, through the sampling scheme given in Section 2.2, the same random graph model as the classical RGG model on $\mathbb{S}^{d-1}$ with threshold 0. Thus two sampled points $X_i, X_j \in \mathbb{S}^{d-1}$ will be connected if and only if they lie in the same semisphere.

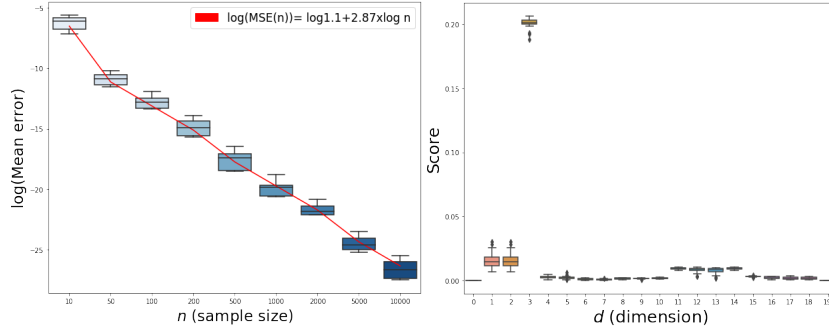

Figure 1: In the left we have a boxplot of $MSE_n$ for different values of $n$. In the right, we plot the score for a set of candidate dimensions $\mathcal{D} = \{1, \cdots, 19\}$. Data were sampled with $W_1$ on $\mathbb{S}^{d-1}$ with $d = 3$.

We consider different values for the sample size $n$ and for each of them we sample 100 Gram matrices in the case $d = 3$ and run the Algorithm 1 for each. We compute each time the mean squared error, defined by

$$MSE_n = \frac{1}{n^2} \|\hat{\mathcal{G}} - \mathcal{G}^*\|_F^2$$

In Figure 1 we put the $MSE_n$ for different values of $n$, showing how $MSE_n$ decrease in terms of $n$. For each $n$, the $MSE_n$ we plot is the mean over the 100 sampled graphs.

### 4.2 Recovering the dimension $d$

We conducted a simulation study using graphon $W_1$, sampling 1000 point on the sphere of dimension $d = 3$ and we use Algorithm 1 to compute a score and recover $d$. We consider a set of candidates with $d_{max} = 15$. In Figure 1 we provide a boxplot for the score of each candidate repeating the procedure 50 times. We see that for this graphon, the algorithm can each time differentiates the true dimension from the "noise".

| $n$(sample size) | runtime(seconds) |
|---|---:|
| 10 | 0.012 |
| 50 | 0.016 |
| 100 | 0.020 |
| 200 | 0.040 |
| 500 | 0.19 |
| 1000 | 1.02 |
| 2000 | 5.07 |
| 5000 | 62.37 |
| 10000 | 450.53 |

Figure 2: The mean (25 repetitions) runtime of the HEiC algorithm for the graphon $W_1$. The experiments were performed on a 3,3Ghz Intel i5 with 16GB RAM.

## 5  Discussion

Although in this paper we have focused on the sphere as the latent metric space, our main result can be extended to other latent space where the distance is translation invariant, such as compact Lie groups or compact symmetric spaces. In that case, the geometric graphon will be of the form $W(x, y) = f(\cos \rho(x, y))$ where $x, y$ are points in the compact Lie group $\mathbb{S}$ and $\rho(\cdot, \cdot)$ is the metric. We will have

$$f(\cos \rho(x, y)) = f(\cos \rho(x \cdot y^{-1}, e_1)) = \tilde{f}(x \cdot y^{-1})$$

where $e_1$ is the identity element in $\mathbb{S}$ and $\tilde{f}(x) = f(\rho(x, e_1))$. In consequence $W(x, y) = \tilde{f}(x \cdot y^{-1})$. Furthermore, there exists an addition theorem in this case (which is central to our recovery result). Analogous regularity notions to the one considered in this work are also worth exploring. In [8] the authors give more details on the model of geometric graphon in compact Lie groups with focus on the graphon estimation.

In principle, it would be possible to extend most of the results of this paper to the case when the underlying space is $\mathbb{B}^d = \{x \in \mathbb{R}^d : \|x\| \le 1\}$ and the link function depends only on the inner products of the points in $\mathbb{B}^d$. As detailed in [7], the harmonic analysis on the sphere can be extended to the unit ball. In particular, an analogous addition theorem exists. Besides, one fundamental fact that used in the proof of Theorem 1 is the control on the growth of the $L^2(\mathbb{S}^{d-1})$ norm of the spherical harmonics, which has its analog for the polynomial base in $L^2(\mathbb{B}^d)$. Despite the similarities between the model on the unit sphere and the model on the unit ball, they might generate very different graphs. For instance, an interesting feature of the model on $\mathbb{B}^d$ is that is not only angle dependent (as in the case of the unit sphere), but also norm dependent. This would allow to generate graphs with more heterogenous node distribution. The study in depth of this model is left for a future work as well as the study of the sparser case.

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
