[Supplementary Material]

# Supplementary Material for Latent Distance Estimation for Random Geometric Graphs

**Ernesto Araya**
Laboratoire de Mathématiques d'Orsay (LMO)
Université Paris-Sud
91405 Orsay, France
`ernesto.araya-valdivia@u-psud.fr`

**Yohann De Castro**
Institut Camille Jordan
École Centrale de Lyon
69134 Écully, France
`yohann.de-castro@ec-lyon.fr`

## 1 Graphon regularity

One way to define the regularity of a geometric graphon on $\mathbb{S}^{d-1}$ is through the notion of weighted Sobolev spaces on the interval $[-1, 1]$. In this context, the regularity is related to the rate at which the eigenvalue sequence $\{\lambda_i^*\}_{i=0}^\infty$ tends to 0. Here we follow [6]. For a function of the form $f(t) = \sum_{k \geq 0} \lambda_k^* c_k G_k^\gamma(t)$, we define the norm

$$\|f\|_{Z_\gamma^s}^2 = \sum_{k=0}^\infty d_k |\lambda_k^*|^2 \big(1 + k(k + 2\gamma + 1)\big)^s$$

We will say that $f$ belongs to weighted Sobolev space $Z_\gamma^s$ if $\|f\|_{Z_\gamma^s} \leq \infty$. We will refer to $s$ as the regularity parameter. As in the case with classical Sobolev spaces, there is a definition of weighted Sobolev spaces that involves the integrability (with respect to the measure $w_\gamma(t)dt$) of the weak derivatives of a function. That is a function $f$ belongs to $Z_\gamma^s$ if it has $s$ weak derivatives that are integrable with respect to the weighted $L^2$ norm in $[-1, 1]$ with weight $w_\gamma(t) = (1 - t)^{\gamma - \frac{1}{2}}$. In [6] the authors prove that both definitions are in fact equivalent.

## 2 Geometric Graphons have $\lambda_0^*$ as the largest eigenvalue

To avoid border issues in HEiC algorithm, we use the fact that the eigenvalue $\lambda_0^*$ associated to the Gegenbauer polynomial $G_0^\gamma(t) = \mathbb{1}(t) := 1$ for $t \in [-1, 1]$ is the largest one, which in the notation of the paper can be written as $\lambda_0^{\text{sort}} = \lambda_0^*$. This is true for all geometric graphons.

**Lemma 1.** *If $W : \mathbb{S}^{d-1} \times \mathbb{S}^{d-1} \to [0, 1]$ is such that*

$$W(x, y) = f(\langle x, y \rangle)$$

*for $f : [-1, 1] \to [0, 1]$, then*

$$d_W(x) := \int_{\mathbb{S}^{d-1}} W(x, y) d\sigma(y)$$

*is constant.*

*Proof.* The proof follows from a change of variable. $\square$

The following theorem is an analogous result to a classical theorem of spectral graph theory

**Theorem 2.** *For a graphon $W : \mathbb{S}^{d-1} \times \mathbb{S}^{d-1} \to [0, 1]$ we have*

$$\int_{\mathbb{S}^{d-1} \times \mathbb{S}^{d-1}} W(x, y) d\sigma(x) d\sigma(y) \leq \lambda_0^* \leq \max_{x \in \mathbb{S}^{d-1}} d(x)$$

*Proof.* By Courant-Fisher min-max principle we have

$$\lambda_0^* = \max_{f \in L^2([-1,1])} \frac{\langle T_W f, f \rangle}{\langle f, f \rangle}$$

In particular, if we take the function $\mathbb{1}(x) := 1$ for $x \in [-1, 1]$ we have

$$\lambda_0^* \geq \frac{\langle T_W \mathbb{1}, \mathbb{1} \rangle}{\langle \mathbb{1}, \mathbb{1} \rangle}$$

$$= \frac{\int_{\mathbb{S}^{d-1}} W(x,y) d\sigma(x) d\sigma(y)}{\int_{\mathbb{S}^{d-1}} d\sigma(y)}$$

$$= d_W$$

the last follows form the definition of $d_W$ and the fact that $\sigma$ is a probability measure on the sphere. On the other hand, if $f_0$ is an eigenfunction associated with $\lambda_0$ we can choose $x^*$ such that $f_0(x^*) \geq f_0(x)$ for $x \in [-1, 1]$. Without loss of generality, assume that $f_0(x^*) \neq 0$. So

$$\lambda_0^* = \frac{T_W f_0(x^*)}{f_0(x^*)}$$

$$= \int_{\mathbb{S}^{d-1}} W(x^*, y) \frac{f_0(y)}{f_0(x^*)} d\sigma(y)$$

$$\leq \int_{\mathbb{S}^{d-1}} W(x^*, y) d\sigma(y)$$

$$= d_W(x^*)$$

which finish the proof $\qquad\square$

Since $G_0^\gamma(t) = \mathbb{1}(t)$ we have by Lemma 1 and Theorem 2 that the $\lambda_0^* = \lambda_0^{\text{sort}}$.

## 3 Proof of the rate of convergence of HEiC Algorithm

This section is devoted to the proof of the main theorem, Theorem 2.2. In the sequel, the sentence "$n$ large enough" means that $n$ is bigger than some $n_0 \geq 1$ that may depend on $W$ and $\alpha$ (we will make this explicit). Recall that the result obtained will hold with probability $1 - \alpha$ with $\alpha > 0$ arbitrarily small. We used through the paper, the notation $X \leq_\alpha C$, where $X$ is random variable and $C$ a constant, to indicate that the inequality holds with probability bigger than $1 - \alpha$.

The aim is to bound $\|\mathcal{G}^* - \hat{\mathcal{G}}\|_F$ and we will split it into two terms as follows

$$\|\mathcal{G}^* - \hat{\mathcal{G}}\|_F \leq \|\mathcal{G}^* - \mathcal{G}\|_F + \|\mathcal{G} - \hat{\mathcal{G}}\|_F$$

where the matrix $\mathcal{G}$ will be defined later (see Proposition 3) using a subset of eigenvectors $V$ of $T_n$. We will treat these terms separately starting with $\|\mathcal{G} - \hat{\mathcal{G}}\|_F$ in Section 3.2 and the other will be treated in Section 3.3.

The first step is to control the probability of the following event $\mathcal{E}$

$$\mathcal{E} := \left\{ \delta_2 \left( \lambda\left(\frac{1}{\rho_n} T_n\right), \lambda(T_W) \right) \vee \frac{2^{\frac{9}{2}} \sqrt{d}}{\rho_n \Delta^*} \|T_n - \hat{T}_n\|_{op} \leq \frac{\Delta^*}{8} \right\},$$

where $\Delta^*$ is the spectral gap $\text{Gap}_1(W)$. We will prove in Section 3.1 that this event holds with probability $1 - \alpha/2$ when $n$ is large enough. This event ensures that the "noise level" is lower than the spectral gap $\text{Gap}_1(W)$ and it guarantees that our algorithm recovers the right subset of eigenvectors as will see in Proposition 3, Section 3.1.

### 3.1 Event guaranteeing the algorithm convergence

Invoke Theorem 7 with $Y = \hat{T}_n - T_n$, which by definition have independent centered entries (conditional to latent points $\{X_i\}_{i=1}^n$), to obtain

$$\mathbb{P}\left( \|\hat{T}_n - T_n\|_{op} \geq \frac{3\sqrt{2D_0}}{n} + C_0 \frac{\sqrt{\log n/\alpha}}{n} \right) \leq \alpha$$

for $\alpha \in (0, 1/3)$. Note that for $n$ large enough, one has

$$\|\hat{T}_n - T_n\|_{op} \leq_{\alpha/4} C \max\Big\{\sqrt{\frac{\rho_n}{n}}, \frac{\sqrt{\log n}}{n}\Big\}$$

by Theorem 7, because $D_0 = \max_{0 \leq i \leq n} \sum_{j=1}^{n} \Theta_{ij}(1 - \Theta_{ij})$ is $\mathcal{O}(n\rho_n)$, by the definition of $\Theta$. Thus, for $n$ large enough we have

$$\frac{1}{\rho_n}\|\hat{T}_n - T_n\|_{op} \leq_{\alpha/4} C \max\Big\{\frac{1}{\sqrt{\rho_n n}}, \frac{\sqrt{\log n}}{\rho_n n}\Big\} \leq \frac{(\Delta^*)^2}{2^{\frac{17}{2}}\sqrt{d}}, \tag{1}$$

provided that $\frac{\sqrt{\log n}}{\rho_n n} = o(1)$, which is the case when $\rho_n = \Omega(\log n/n)$, which we have called the *relatively sparse* case. Let $V \in \mathbb{R}^{n \times d}$ and $\hat{V} \in \mathbb{R}^{n \times d}$ be two matrices with columns corresponding to the eigenvectors associated to eigenvalues $\lambda_{i_1}, \lambda_{i_2}, \ldots, \lambda_{i_d}$ and $\hat{\lambda}_{i_1}, \hat{\lambda}_{i_2}, \cdots, \hat{\lambda}_{i_d}$ of $T_n$ and $\hat{T}_n$ respectively, as in Theorem 8. We use Lemma 5 with $A = \hat{V}\hat{O}$ and $B = V$, where $\hat{O}$ is an orthogonal matrix, and Theorem 8 assuming that the right hand side of (7) is smaller than 1, obtaining

$$\|\hat{V}\hat{V}^T - VV^T\|_F \leq 2\|\hat{V}\hat{O} - V\|_F$$
$$\leq \frac{2^{\frac{5}{2}}\min\{\sqrt{d}\|T_n - \hat{T}_n\|_{op}, \|T_n - \hat{T}_n\|_F\}}{\Delta} \tag{2}$$

where $\Delta := dist(\{\lambda_{i_1}, \cdots, \lambda_{i_d}\}, \lambda(T_n) \setminus \{\lambda_{i_1}, \cdots, \lambda_{i_d}\})$. Then we have

$$\|\hat{V}\hat{V}^T - VV^T\|_F \leq \frac{2^{\frac{5}{2}}\frac{\sqrt{d}}{\rho_n}\|T_n - \hat{T}_n\|_{op}}{\frac{1}{\rho_n}\Delta}$$
$$\leq_\alpha \frac{\rho_n(\Delta^*)^2}{2^6\Delta} \tag{3}$$

Now, we use the $\delta_2$ metric to quantify the convergence of the eigenvalues of the normalized probability matrix $\frac{1}{\rho_n}T_n$ to the eigenvalues of the integral operator $T_W$. From Theorem 14 we have that, when $n$ is large enough

$$\delta_2\Big(\lambda(\frac{1}{\rho_n}T_n), \lambda(T_W)\Big) \leq_{\alpha/4} C\Big(\frac{\log n}{n}\Big)^{\frac{s}{2s+d-1}} \leq \frac{\Delta^*}{8}, \tag{4}$$

where $\Delta^*$ is the spectral gap $\mathrm{Gap}_1(W)$. This and (1) ensure that $\mathcal{E}$ has probability $1 - \alpha/2$. In particular, it gives the following result proving that our algorithm find the right eigenvectors.

**Proposition 3.** *On the event $\mathcal{E}$, there exists one and only one set $\Lambda_1$ of $d$ eigenvalues of $\frac{1}{\rho_n}\hat{T}_n$ separated by at least $\Delta^*/2$ from the other eigenvalues of $\hat{T}_n$. These eigenvalues are at a distance at most $\Delta^*/8$ of $\frac{1}{\rho_n}\lambda_1, \ldots, \frac{1}{\rho_n}\lambda_d$, the eigenvalues of $T_n$ whose eigenvectors define the matrix $\mathcal{G} := (1/c_1)VV^T$. Furthermore, on the event $\mathcal{E}$, our algorithm returns the matrix $\hat{\mathcal{G}} = (1/c_1)\hat{V}\hat{V}^T$ composed by the eigenvectors corresponding to the eigenvalues of $\Lambda_1$.*

*Proof.* When $\Delta^* > 0$, we remark that $\lambda_1^* = \lambda_2^* = \ldots = \lambda_d^*$ is the only eigenvalue of $T_W$ with multiplicity $d_1 = d$, the others eigenvalues (except for $\lambda_0^*$) having multiplicity strictly greater than $d$. Now, using (4) we deduce that there exists a unique set $\frac{1}{\rho_n}\lambda_{i_1}, \frac{1}{\rho_n}\lambda_{i_2}, \ldots, \frac{1}{\rho_n}\lambda_{i_d}$ of $d$ eigenvalues of $T_n$ that can be separated from the other eigenvalues by a distance at least $3\Delta^*/4$, namely the triangular inequality gives

$$\frac{\Delta}{\rho_n} \geq \frac{3\Delta^*}{4}. \tag{5}$$

To these eigenvalues correspond the eigenvectors $V \in \mathbb{R}^{n \times d}$ defining $\mathcal{G} := (1/c_1)VV^T$.

Furthermore, using (3) we get that there exists eigenvalues $\hat{\lambda}_{i_1}, \hat{\lambda}_{i_2}, \ldots, \hat{\lambda}_{i_d}$ and eigenvectors $\hat{V} \in \mathbb{R}^{n \times d}$ of $\hat{T}_n$ such that $\|\hat{V}\hat{V}^T - VV^T\|_F \leq \Delta^*/48$. We define $\Lambda_1 := \{\hat{\lambda}_{i_1}, \cdots, \hat{\lambda}_{i_d}\}$. By Hoffman-Wielandt inequality [2, Thm.VI.4.1], it holds

$$\Big(\sum_{k=1}^{d}(\hat{\lambda}_k^{\mathrm{sort}} - \lambda_k^{\mathrm{sort}})^2\Big)^{1/2} \leq \|\hat{V}\hat{V}^T - VV^T\|_F \leq \Delta^*/8,$$

where $\hat{\lambda}_1^{\mathrm{sort}} \geq \hat{\lambda}_2^{\mathrm{sort}} \geq \cdots \geq \hat{\lambda}_d^{\mathrm{sort}}$ (resp. $\lambda_1^{\mathrm{sort}} \geq \lambda_2^{\mathrm{sort}} \geq \cdots \geq \lambda_d^{\mathrm{sort}}$) is the sorted version of the eigenvalues $\hat{\lambda}_{i_1}, \cdots, \hat{\lambda}_{i_d}$ (resp. $\lambda_{i_1}, \cdots, \lambda_{i_d}$). By triangular inequality, we deduce that

$$\hat{\Delta} := dist(\Lambda_1, \lambda(\hat{T}_n) \setminus \Lambda_1) \geq \frac{\Delta^*}{2} \,,$$

namely $\hat{\lambda}_{i_1}, \hat{\lambda}_{i_2}, \ldots, \hat{\lambda}_{i_d}$ is a set of $d$ eigenvalues at distance at least $\Delta^*/2$ from the other eigenvalues of $\hat{T}_n$.

This analysis can be also done for the other eigenvalues as follows. Eq. (4) shows that there exists a set of $d_k$ eigenvalues of $T_n$ which concentrate around $\mu_k^*$, and such that it has diameter smaller than $\Delta^*/4$. Recall that $d_k$ is the size of the Spherical Harmonics space $k$ and $d_k > d_1 = d$. Weyl's inequality [2, P.63] shows that there exists a set $\Lambda_k$ of $d_k$ eigenvalues of $\hat{T}_n$ around $\mu_k^*$ of size $\Delta^*/4$. Now, consider a subset $L$ of $d$ eigenvalues which is different from $\Lambda_1$ then the previous discussion shows that there exists an eigenvalue $\hat{\lambda}$ which is not in $L$ and that belongs to same cluster to one of the eigenvalues in $L$. In particular $\hat{\lambda}$ is at a distance less than $\Delta^*/4$ of $L$. By (5) we deduce that, on the event $\mathcal{E}$, Algorithm 1 returns $\hat{\mathcal{G}} = (1/c_1)\hat{V}\hat{V}^T$ composed by the eigenvectors corresponding to the eigenvalues of the aforementioned cluster of $d$ eigenvalues. $\qquad\square$

We now prove the following lemma, which is stated in the article

**Lemma 4.** *On the event $\mathcal{E}$, the following equality holds*

$$\mathrm{Gap}_1(\hat{T}_n) = \max \left\{ \max_{1 \leq i \leq n-d} \min \left\{ \mathrm{left}(i), \mathrm{right}(i+d) \right\}, \mathrm{left}(n-d+1) \right\}$$

*Proof.* The lemma follows from Proposition 3. Indeed, on the event $\mathcal{E}$ there exist only one set $\Lambda_1$ of eigenvalues of $\hat{T}_n$ with cardinality $d$, whose distance to the rest of the spectrum is larger that $\Delta^*$ and its diameter is smaller that $\Delta^*$. When sorting the eigenvalues of $\hat{T}_n$ in decreasing order, those belonging to $\Lambda_1$ will appear in consecutive order. The lemma follows from this observation and from the fact $\mathrm{Gap}_1(\hat{T}_n; i_{n-d-1}, \cdots, i_{n-1}) = \mathrm{left}(n-d-1)$. $\qquad\square$

### 3.2 Sampling error control

We have by (2) that

$$\|\hat{\mathcal{G}} - \mathcal{G}\|_F = \frac{1}{c_1}\|\hat{V}\hat{V}^T - VV^T\|_F \leq_\alpha C\frac{(d-2)}{\sqrt{dn}} \,, \tag{6}$$

whenever $n$ is large enough and $\Delta^* > 0$, where $C$ may depend on $W$. In the last inequality we used that $c_1 = d/(d-2)$.

### 3.3 Sampled eigenvectors convergence

We are left to control $\|\mathcal{G}^* - \mathcal{G}\|_F$. We begin by recalling some basic definitions we have made through the paper and introducing some notation. Set $R = \mathcal{O}((n/\log n)^{\frac{1}{2s+d-1}})$ and $\tilde{R} := d_0 + d_1 + \ldots + d_R$ the total size of the $R+1$ first Harmonic spaces. It is well known that $\tilde{R} = \mathcal{O}(R^{d-1}) = o(n)$ for $s > 0$. If $W_R$ is the rank $R'$ approximation of $W$, we have

$$T_R = \left(\frac{1}{n}W_R(X_i, X_j)\right)_{i,j} = \Phi_{0,R}\Lambda_{0,R}^*\Phi_{0,R}$$

where $\Phi_{0,R}$ is the matrix with columns $\Phi_k \in \mathbb{R}^n$, for $0 \leq k \leq R'$, such that $(\Phi_k)_i = \phi_k(X_i)$ and $\Lambda_{0,R}^* = diag(\lambda_0^*, \lambda_2^*, \cdots, \lambda_{\tilde{R}}^*)$. Similarly $\Lambda_{0,R} = diag(\lambda_0, \lambda_2, \cdots, \lambda_{\tilde{R}})$. Let $\tilde{V}$ be the matrix that contains as columns the eigenvectors of the matrix $T_n$ and $\tilde{V}_R$ contains as columns the eigenvectors $T_R$ so we have the eigenvalue decomposition

$$T_n = \tilde{V}\Lambda\tilde{V}^T$$

$$T_R = \tilde{V}_R\Lambda_R\tilde{V}_R^T$$

Let $V$ be the matrix that contains the columns $1, \cdots, d$ of $\tilde{V}$, $V_R$ contains the columns $1, \cdots, d$ of $\tilde{V}_R$ and $V^*$ contains $\phi_k$ for $1 \le k \le d$ as columns. Then $\mathcal{G}^*, \mathcal{G}, \mathcal{G}_R, \mathcal{G}_{proj}^*$ are defined by

$$\mathcal{G}^* := \frac{1}{c_1} V^* (V^*)^T$$

$$\mathcal{G} := \frac{1}{c_1} V V^T$$

$$\mathcal{G}_R := \frac{1}{c_1} V_R V_R{}^T$$

$$\mathcal{G}_{proj}^* := V^* (V^{*T} V^*)^{-1} V^{*T}$$

Note that $\mathcal{G}_{proj}^*$ is the projection matrix for the column span of the matrix $V^*$, that is, it is the projection matrix onto the space $\operatorname{span}\{\Phi_1, \cdots, \Phi_d\}$.

We have by triangle inequality

$$\|\mathcal{G}^* - \mathcal{G}\|_F \le \|\mathcal{G}^* - \mathcal{G}_{proj}^*\|_F + \|\mathcal{G}_{proj}^* - \mathcal{G}_R\|_F + \|\mathcal{G}_R - \mathcal{G}\|_F$$

We call truncation error to the last term in the right hand side, because it is related to the fact that $W_R$ is a rank $R'$ approximation of $W$.

To bound $\|\mathcal{G} - \mathcal{G}_R\|_F$ we will use Theorem 8 noting that $\mathcal{G}$ and $\mathcal{G}_R$ have as columns the eigenvectors of matrices $T_n$ and $T_R$. So

$$\|\mathcal{G} - \mathcal{G}_R\|_F \le \frac{2^{\frac{3}{2}} \|T_n - T_R\|_F}{\Delta} \le C \frac{(n/\log n)^{-s/(2s+d-1)}}{\Delta}$$

where we recall that $R = \mathcal{O}((n/\log n)^{\frac{1}{2s+d-1}})$. In order to bound $\|\mathcal{G}^* - \mathcal{G}_{proj}^*\|_F$ we use Lemma 6 with $B = V^*$ obtaining

$$\|\mathcal{G}^* - \mathcal{G}_{proj}^*\|_F \le \|\operatorname{Id}_d - V^{*T} V^*\|_F$$

On the other hand, we have

$$\|\operatorname{Id}_d - V^{*T} V^*\|_F \le \sqrt{d} \|\operatorname{Id}_d - V^{*T} V^*\|_{op}$$

$$\le_\alpha \frac{d}{\sqrt{n}}$$

where we used Theorem 15 to obtain the last inequality.

It only remains to bound the term $\|\mathcal{G}_{proj}^* - \mathcal{G}_R\|_F$. We concentrate first in bounding the term $\mathcal{G}_{proj}^* \mathcal{G}_R^\perp$. We use Theorem 10, with $E = \mathcal{G}_{proj}^*$, $F = \mathcal{G}_R^\perp$, $B = T_R$ and $A = T_R + H$, where

$$H := \tilde{\Phi}_{0,R} \Lambda_{0,R}^* \tilde{\Phi}_{0,R}^T - \Phi_{0,R} \Lambda_{0,R}^* \Phi_{0,R}$$

the matrix $\tilde{\Phi}_{0,R}$ has column $\tilde{\Phi}_k$ for $k \in \{1, \cdots, R'\}$ where the $\tilde{\Phi}_k$ are obtained from $\Phi_k$ by a Gram-Schmidt orthonormalization process. In other words, there exists a matrix $L$ such that $\tilde{\Phi}_{0,R} = \Phi_{0,R}(L^{-1})^T$. The matrix $L$ comes from the Cholesky decomposition of $\Phi_{0,R}^T \Phi_{0,R}$, that is, $L$ satisfy $\Phi_{0,R}^T \Phi_{0,R} = LL^T$.

Note that $A$ and $B$ are symmetric, hence normal matrices, so Theorem 10 applies. Also, in the event $\mathcal{E}$, we can take $S_1 = (\lambda_1 - \frac{\Delta^*}{8}, \lambda_1 + \frac{\Delta^*}{8})$ and $S_2 = \mathbb{R} \setminus (\lambda_1 - \frac{7\Delta^*}{8}, \lambda_1 + \frac{7\Delta^*}{8}))$. By Theorem 10 we have

$$\|\mathcal{G}_{proj}^* \mathcal{G}_R^\perp\|_F \le \frac{\|A - B\|_F}{\Delta^*} = \frac{\|H\|_F}{\Delta^*}$$

where $\Delta := \min_{k,\ell \ne 1,...,d} \{|\lambda_k^* - \lambda_1^*|, |\lambda_d^* - \lambda_\ell^*|\}$. It remains to bound $H$.

We have that

$$\|H\|_F \le \|L^{-T} \Lambda_{0,R}^* L^{-1} - \Lambda_{0,R}^*\|_F \|\Phi_{0,R}^T \Phi_{0,R}\|_{op}$$

$$\le \|\Lambda_{0,R}^*\|_F \|L^{-1} L^{-T} - \operatorname{Id}_{R'}\|_{op} \|\Phi_{0,R}^T \Phi_{0,R}\|_{op}$$

where in the last line we used Corollary 12. It is easy to see that

$$\|L^{-1}L^{-T} - \mathrm{Id}_{R'}\|_{op} = \|(\Phi_{0,R}^T \Phi_{0,R})^{-1} - \mathrm{Id}_{R'}\|_{op}$$

which, using [4, Lem.12], implies that

$$\|Z\|_F \leq_{\alpha/4} 2C_1 \frac{R^{d-1}}{\sqrt{n}}$$

which, since $R = \mathcal{O}((n/\log n)^{\frac{1}{2s+d-1}})$, becomes

$$\|Z\|_F \leq_{\alpha/4} C'\Big(\frac{\log n}{n}\Big)^{\frac{s}{2s+d-1}}$$

for a constant $C' > 0$. Collecting terms we obtain

$$\|\mathcal{G}_{proj}^* \mathcal{G}_R^\perp\|_F \leq_{\alpha/4} \frac{C''}{\Delta^*}\Big(\frac{\log n}{n}\Big)^{\frac{s}{2s+d-1}}$$

Since $\mathcal{G}_{proj}^*$ and $\mathcal{G}_R$ are projectors we have, see [2, p.202]

$$\|\mathcal{G}_{proj}^* - \mathcal{G}_R\|_F = 2\|\mathcal{G}_{proj}^* \mathcal{G}_R^\perp\|_F$$

which implies that

$$\|\mathcal{G}_{proj}^* - \mathcal{G}_R\|_F \leq_{\alpha/4} \frac{2C''}{\Delta^*}\Big(\frac{n}{\log n}\Big)^{\frac{-s}{2s+d-1}}$$

To conclude, we have that

$$\|\mathrm{Id}_d - V^{*T}V^*\|_F \leq \sqrt{d}\,\|\mathrm{Id}_d - V^{*T}V^*\|_{op}$$
$$\leq_{\alpha/4} \frac{d}{\sqrt{n}}$$

where we use Theorem 15 in the second inequality. Collecting terms we conclude that

$$\|\mathcal{G}^* - \mathcal{G}\|_F \leq_{\alpha/4} \frac{C_d}{\Delta^*}\Big(\frac{\log n}{n}\Big)^{\frac{s}{2s+d-1}}$$

where $C_d$ is a constant that depends on $d$ and $\alpha$.

## 4 Useful results

**Lemma 5.** *Let $A$, $B$ be two matrices in $\mathbb{R}^{n\times d}$ then*

$$\|AA^T - BB^T\|_F \leq (\|A\|_{op} + \|B\|_{op})\|A - B\|_F$$
$$\|AA^T - BB^T\|_{op} \leq (\|A\|_{op} + \|B\|_{op})\|A - B\|_{op}\,.$$

*If it holds that $A^T A = B^T B = I_d$ then*

$$\|AA^T - BB^T\|_F \leq 2\|A - B\|_F$$

*Proof.* We begin with the first inequality

$$\|AA^T - BB^T\|_F = \|(A - B)A^T + B(A^T - B^T)\|_F$$
$$\leq \|(A \otimes I_n)\mathrm{vec}(A - B)\|_2 + \|(I_d \otimes B)\mathrm{vec}(A - B)^T\|_2$$
$$\leq (\|A \otimes I_n\|_{op} + \|I_d \otimes B\|_{op})\|A - B\|_F$$
$$= (\|A\|_{op} + \|B\|_{op})\|A - B\|_F\,.$$

Here $\mathrm{vec}(\cdot)$ represent the vectorization of a matrix, that its transformation into a column vector. The second inequality is given by

$$\|AA^T - BB^T\|_{op} = \|(A - B)A^T + B(A^T - B^T)\|_{op}$$
$$\leq (\|A\|_{op} + \|B\|_{op})\|A - B\|_{op}\,.$$

The third statement is an elementary consequence of the above inequalities. □

**Lemma 6.** *Let $B$ a $n \times d$ matrix with full column rank. Then we have*

$$\|BB^T - B(B^TB)^{-1}B^T\|_F = \|\mathrm{Id}_d - B^TB\|_F$$

*Proof.* We have

$$\|BB^T - B(B^TB)^{-1}B^T\|_F = \|B\big((B^TB)^{-1} - \mathrm{Id}_d\big)B^T\|_F$$

and by definition of the Frobenious norm and cyclic property of the trace

$$\|B\big((B^TB)^{-1} - \mathrm{Id}_d\big)B^T\|_F^2 = tr\big(B((B^TB)^{-1} - \mathrm{Id}_d)B^TB((B^TB)^{-1} - \mathrm{Id}_d)B^T\big)$$
$$= tr\big((\mathrm{Id}_d - B^TB)^2\big)$$
$$= \|\mathrm{Id}_d - B^TB\|_F^2$$

$\square$

## 4.1 Bandeira-Van Handel theorem

The following theorem is a slight reformulation of the [1, Cor.3.12]

**Theorem 7** (Bandeira-Van Handel). *Let $Y$ be a $n \times n$ symmetric random matrix whose entries $Y_{ij}$ are independent centered random variables. There exists a universal constant $C_0$ such that for $\alpha \in (0,1)$*

$$\mathbb{P}\Big(\|Y\|_{op} \geq 3\sqrt{2D_0} + C_0\sqrt{\log n/\alpha}\Big) \leq \alpha$$

*where $D_0 = \max_{0 \leq i \leq n} \sum_{j=1}^n Y_{ij}(1 - Y_{ij})$.*

*Proof.* By [1, Rmk.3.13] we have the tail concentration bound (taking their $\epsilon$ equal to $1/2$)

$$\mathbb{P}\Big(\|Y\|_{op}\Big) \geq 3\sqrt{2D_0} + \max_{ij}|Y_{ij}|C_0\sqrt{\log n/\alpha}$$

the result follows, because $\max_{ij}|Y_{ij}| \leq 1$. $\square$

Using the previous theorem with $Y = \hat{T}_n - T_n$, which is centered and symmetric, we obtain the tail bound

$$\mathbb{P}\Big(\|\hat{T}_n - T_n\|_{op} \geq \frac{3\sqrt{2D_0}}{n} + C_0\frac{\sqrt{\log n/\alpha}}{n}\Big) \leq \alpha$$

## 4.2 Davis-Kahan $\sin\theta$ theorem

For $n$ large enough, the eigenspace associated to the eigenvalue $\hat{\lambda}_1$ is close to the eigenspace associated to the eigenvalue $\lambda_1$. This is precised by the Davis-Kahan $\sin\theta$ theorem. We use the following version which is proved in [9]

**Theorem 8.** *Let $\Sigma$ and $\hat{\Sigma}$ be two symmetric $\mathbb{R}^{n \times n}$ matrices with eigenvalues $\lambda_1 \geq \lambda_2 \geq \cdots \geq \lambda_n$ and $\hat{\lambda}_1 \geq \hat{\lambda}_2 \geq \cdots \hat{\lambda}_n$ respectively. For $1 \leq r \leq s \leq n$ fixed, we assume that $\min\{\lambda_{r-1} - \lambda_r, \lambda_s - \lambda_{s-1}\} > 0$ where $\lambda_0 := \infty$ and $\lambda_{n+1} = -\infty$. Let $d = s - r + 1$ and $V$ and $\hat{V}$ two matrices in $\mathbb{R}^{n \times d}$ with columns $(v_r, v_{r+1}, \cdots, v_s)$ and $(\hat{v}_r, \hat{v}_{r+1}, \cdots, \hat{v}_s)$ respectively, such that $\Sigma v_j = \lambda_j v_j$ and $\hat{\Sigma}\hat{v}_j = \hat{\lambda}_j\hat{v}_j$. Then there exists an orthogonal matrix $\hat{O}$ in $\mathbb{R}^{d \times d}$ such that*

$$\|\hat{V}\hat{O} - V\|_F \leq \frac{2^{3/2}\min\{\sqrt{d}\|\Sigma - \hat{\Sigma}\|_{op}, \|\Sigma - \hat{\Sigma}\|_F\}}{\min\{\lambda_{r-1} - \lambda_r, \lambda_s - \lambda_{s+1}\}} \tag{7}$$

Also, we need the following perturbation result [2, Thm.VII.2.8]

**Theorem 9.** *Let $A$ and $B$ two the normal matrices and define $\delta = dist(\lambda(A), \lambda(B))$. If $X$ satisfies the Sylvester equation $AX - XB = Y$, then*

$$\|X\|_F \leq \frac{1}{\delta}\|Y\|_F$$

Another useful perturbation theorem [2, Thm.VII.3.1]

**Theorem 10.** *Let $A$ and $B$ be two normal operators and $S_1$ and $S_2$ two sets separated by a strip of size $\delta$. Let $E$ be the orthogonal projection matrix of the eigenspaces of $A$ with eigenvalues inside $S_1$ and $F$ be the orthogonal projection matrix of the eigenspaces of $B$ with eigenvalues inside $S_2$. Then*

$$\|EF\|_F \leq \frac{1}{\delta}\|E(A-B)F\|_F \leq \frac{1}{\delta}\|A-B\|_F$$

### 4.3 Ostrowski theorem

The following eigenvalue perturbation theorem is due to Ostrowski [5, Thm.4.5.9] and [3, Cor.3.54]

**Theorem 11.** *Let $A \in \mathbb{R}^{n \times n}$ be a Hermitian matrix and $S \in \mathbb{R}^{n \times n}$ be a nonsingular matrix. Then for each $1 \leq i \leq n$ there exists $\theta_i > 0$ such that*

$$\lambda_i(SAS^*) = \theta_i \lambda_i(A)$$

*In addition, it holds*

$$|\lambda_i(SAS^*) - \lambda_i(A)| \leq |\lambda_i(A)| \|S^*S - \mathrm{Id_n}\|_{op}$$

**Remark 1.** *The previous theorem is also valid for $S$ singular [5, Cor.4.5.11].*

The previous theorem can be extended to the case where $S$ is not necessarily a square matrix [3, Cor.3.59]

**Corollary 12.** *Let $A \in \mathbb{R}^{n \times n}$ be a Hermitian matrix and $S \in \mathbb{R}^{d \times n}$ matrix then*

$$|\lambda_i(SAS^*) - \lambda_i(A)| \leq |\lambda_i(A)| \|S^*S - \mathrm{Id_n}\|_{op}$$

From the previous result we deduce the following corollary

**Corollary 13.** *Under the same conditions of Corollary 12 we have*

$$\|SAS^* - A\|_F \leq \|A\|_F \|S^*S - \mathrm{Id_n}\|_{op}$$

### 4.4 Convergence rate of regular graphon estimation

We use the following result, which can be found in [4]

**Theorem 14.** *Let $W$ be a graphon on the sphere of the form $W(x,y) = f(\langle x, y \rangle)$. If $f$ belongs to the weighted Sobolev space $Z^s_{w_\gamma}((-1,1))$ then we have*

$$\delta_2(\lambda(\frac{1}{\rho_n}T_n), \lambda(T_W)) \leq_\alpha C\Big(\frac{\log n}{n}\Big)^{\frac{s}{2s+d-1}}$$

*where $\leq_\alpha$ means that the inequality holds with probability greater than $1 - \alpha$ for $\alpha \in (0, 1/3)$ and $n$ large enough.*

While Theorem 7 gives a bound for the difference of the eigenvalues of the observed matrix with respect to the eigenvalues of the probability matrix, Proposition 14 ensures that the eigenvalues of the empirical matrix are close to these of the integral operator.

### 4.5 Covariance matrix approximation

Given a set of independent random vectors $X_1, \cdots, X_n$ uniformly distributed on the sphere $\mathbb{S}^{d-1}$ we are interested in the concentration properties of the quantity $\frac{1}{n}\sum_{k=1}^n X_i X_i^T$ around its mean, which is $\mathbb{E}(X_i X_i^T) = \mathrm{Id}_d$ for $1 \leq i \leq n$ (in other words, the vectors $X_i$ are isotropic). Since the uniform distribution on the sphere is sub-gaussian [8, Thm.3.4.6], we can use the following theorem [7, Prop.2.1].

**Theorem 15.** *If $X_1, \cdots, X_n$ are independent random vectors in $\mathbb{R}^d$ with $d \leq n$ which have sub-gaussian distribution. Then for any $\alpha \in (0,1)$ it holds*

$$\Big\|\frac{1}{n}\sum_{k=1}^n X_k X_k^T - \mathrm{Id}_d\Big\|_{op} \leq_\alpha \sqrt{\frac{d}{n}}$$