[Reviews · NeurIPS 2019]

Reviewer 1



Originality The paper seems fairly original. The idea of leveraging known facts about linear eigenfunctions to obtain a relation between the underlying Gram matrix and the spectrum of the adjacency matrix is nice. Based on my reading of the supplement, the analysis rests heavily on prior work which bounds the error between the spectrum of T_W and T_n, in the case where W is Sobolev smooth. Quality From a limited reading of the supplement, the proof techniques make nice use of a variety of facts in operator perturbation theory, and seem interesting independent of this application. Simulations could be improved. For instance, the left plot in figure 1 could include a slope, to demonstrate the error is obeying the stated rate in Theorem 4. The probability of event E holding ‘for n large enough’ seems weak, particularly since Theorem 4 claims an asymptotic rate of estimation. Could a sense be given of how P(E) depends on n and W? Some typos, including [line number]: [32]: ‘mane’ [58]: ‘though as’ [76]: ‘nonassymptotic’ [81]: ‘UVST’ [148]: ‘need precise eigenvalue...’ [208]: ‘versin’ [224]: [Algorithm 1]: Lambda_1 is initialized to depend on lambda_i before i is defined. Should this be lambda_1? Clarity The role various elements play in the analysis could be more explicitly untangled. For instance, if I understand correctly, the Sobolev rate comes from the estimation of the spectrum of the operator T_W by the (unobserved) operator T_n, which dominates the rate of estimation of the spectrum of the operator T_n by \hat{T})_n. Similarly, given the primarily theoretical nature of this paper, more detail should be given regarding the techniques used for novel aspects of proof. For instance, on lines [78-80], the authors mention “adaption of perturbation theorems for matrix projection… to the ‘nearly orthogonal’ case.” It might have been nice to expound on this in the main text. Significance The setup is limited to the case where latent variables are sampled uniformly over the sphere. This seems unrealistic in practice and is additionally unverifiable, considering the variables are latent. The discussion section mentions extensions, but they are still restrictive, as in general the analysis critically rests on the ‘addition theorem’. Overall, this work therefore is of theoretical interest but seems unlikely to have practical impact. Particularly in light of these practical limitations, more motivation for the specific problem of latent distance estimation should have been given.

Reviewer 2



The theory presented in this paper is solid and can provide basis for further research in this area, nevertheless the experiments section can use more exploration. Scalability is not mentioned for instance. It would have been helpful to understand how does this relate to real world graphs as well. Upon looking through the author(s) response, my confidence is increased in the theoretical findings of this paper, and its ability to serve as a building block towards more complex models -- but I still think a more extensive experiments section could make the paper much stronger.

Reviewer 3



The authors consider the problem of estimating the latent variables of graphon defined on a sphere. They introduce a spectral algorithm to estimate the pairwise distances between the latent points and prove its rate of convergence. Originality. ============ The paper contains original results about the estimation of the underlying pairwise distancaes oof the latent points of a graphon (defined on a sphere). A lot of the mathematical machinery developed draws from the existing literature, but given the technical nature of the topic this is to be expected. The results are supported by theory and a set of numerical experiments. Clarity. ======== The paper is written in a clear and coherent manner. One question I have about the numerical experiments is why the score function appears to be peak not only at dimension three, but seems to have some further local maxima at 7, 11, etc. Is there an intuitive reason for this? Significance ============ Graphon estimation and latent variable models for graphs are of great relevance for learning from graphical data. Hence the contribution of the authors is certainly relevant. Where the authors could have done a better job is in pointing out potential applications of this kind of model. A brief brief example application to a real-world data set would also have strengthened the paper in my opinion. Quality ======= The quality of the paper is good, overall. There are a few statements discussion items, however, which I find could be improved. The authors state thet the usual embedding space is a sphere or the unit cube in the introduction -- I would think that for geometric graphs the cube is far more common. Are they referring to random dot-product models here? Could they provide some further reference for geometric graphs on the sphere? The authors state that random dot product are more restrictive than the model they consider here, but do not discuss how/why this is the case, so the reader is left hanging. To improve the paper I suggest adding a discussion to make the relationships clear. Are they claiming that their model is a strict superset of RDPGs? If yes please elaborate. Given that their model is based on 'relatively sparse' graphs, but many real-world networks are, in fact, very sparse, I think this aspect (which is a common sticking point with the graphon model) should be discussed in a bit more detail. Do the authors have an idea of how to make their results applicable for sparse graphs (e.g. using graphex or other constructions)? Would there be a problem with the spherical embedding space in this case? Typos: l.208 versin -> version Supplementary information l.226 -- there is a sentence starting with "In Figure.." that does not have an end. Conclusion ========== Overall I think this is a good contribution that would merit publication. [post discussion] After reading the authors' responses and the discussion, I feel this would make an interesting contribution, provided the authors improve the paper along the lines indicated in the response letter.

[Author Response · NeurIPS 2019]

1. **Author response for NeurIPS submission 4700 (Latent distance estimation for random geometric graphs)**

2. We are very grateful to all three reviewers for their time, valuable feedback and suggestions. We highly appreciate the
3. encouraging comments regarding the novelty and solid mathematical analysis of our approach.

4. **I. Motivations and related work.** We agree with the reviewers that more motivation on the spherical setting would
5. strengthen our paper. The model on the sphere has received attention lately, see for example [1] and references therein.
6. One of our contributions is to point out that the spectrum of these graphs is highly structured, which it may have been
7. unnoticed, and to give a method to recover the distances based on this fact. Also, our work may serve to identify the
8. presence of a geometric representation (spherical) by looking at the spectrum of the graph. In terms of modelling, as
9. noted in [1] the sphere would be an appropriate embedding space when each coordinate (feature) of a given point have
10. the same importance in the determination of the geometric representation.

11. Reviewer 3 raised the question of the RDPG model. In general, RDPG model considers latent points $\{X_i\}_{i=1}^n$ and the
12. connection probability is a scaled version of $\langle X_i, X_j \rangle$. In our setting, it corresponds to the link function $f(t) = \frac{1}{2}(1+t)$.

13. **II. Analisys.** Reviewer 1 pointed out that the event $\mathcal{E}$ "holding 'for n large enough" may "seem week". One can derive
14. an explicit bound on $n$ using equation $(1)$ in Sec. 3.1 of the supplementary material. We get that is sufficient that:

$$\max\{\sqrt{\frac{\rho_n}{n}}, \frac{\sqrt{\log n}}{n}\} \leq \frac{\Delta^{*2}}{2^{15/2}C\sqrt{d}} \quad \text{and} \quad \frac{\log n}{n} \leq \left(\frac{\Delta^*}{8C'}\right)^{\frac{2s+d-1}{s}}$$

15. where $C, C' > 0$. We agree that this would shed light on the relation between $n$, $\rho_n$ and the parameters $\Delta^*$, $d$ and $s$.

16. As Rev. 1 mentions one of our contribution is the adaption of matrix perturbation results to a "nearly" orthogonal case,
17. which is detailed in Sec.3 of the supplementary material. Also, it is correct that the Sobolev rate comes mainly by
18. spectral approximation of $T_W$ by $T_n$. We agree that to explicit both points on the main paper will be useful.

19. **III.Experiments.** As suggested by Rev. 1, we include the boxplot for $MSE_n$ accompanied with a curve of the form
20. $MSE_n = Cn^{-r}$ where $r$ is the rate. Here we have a rate $r = 2.87$ for $MSE_n = \frac{1}{n^2}\|\hat{\mathcal{G}} - \mathcal{G}^*\|_F^2$.

21. Rev. 3 asks about an intuitive explanation for the local maxima in the score function, in the dimension recovery method.
22. Given that $d = 3$ the eigenvalue multiplicities are $1, 3, 5, 7, \cdots, 2k+1, \cdots$ for $k \in \mathbb{N}$, thus is not forbidden that the
23. score peaks at any of those values or at a sum of them (meaning that the corresponding eigenvalues are very close).
24. Also, we found a typo in our code and redo the score boxplot for $n = 2000$. The first two figures will replace Fig. 1 of
25. the main paper. In addition, we include the mean (25 rep.) runtime of HEiC alg. for different values of $n$ and correct
26. the typo in the HEiC alg. description. Given that HEiC is spectral algorithm, it will scale roughly as $n^3$.

| $n$(sample size) | runtime(seconds) |
|---|---|
| 10 | 0.012 |
| 50 | 0.016 |
| 100 | 0.020 |
| 200 | 0.040 |
| 500 | 0.19 |
| 1000 | 1.02 |
| 2000 | 5.07 |
| 5000 | 62.37 |
| 10000 | 450.53 |

P.C specs: 3,3GHz Intel i5, 16 GB RAM

**Algorithm 1** Harmonic EigenCluster(HEiC) algorithm

**Input:** $(\hat{T}_n, d)$ adjacency matrix and sphere dimension
$\Lambda^{\text{sort}} = \{\hat{\lambda}_1^{\text{sort}}, \cdots, \hat{\lambda}_{n-1}^{\text{sort}}\} \leftarrow$ eigenvalues of $\hat{T}_n$ sorted in decreasing order
$\Lambda_1 \leftarrow \{\Lambda_1^{\text{sort}}, \cdots, \Lambda_{1+d}^{\text{sort}}\}$: where $\Lambda_i^{\text{sort}}$ is the $i$-th element in $\Lambda^{\text{sort}}$
Initialize $i = 2$, $gap = Gap_1(\hat{T}_n; 1, 2, \cdots, d)$
**while** $i \leq n - d$ **do**
  **if** $Gap_1(\hat{T}_n; i, i+1, \cdots, i+d) > gap$ **then**
    $\Lambda_1 \leftarrow \{\Lambda_i^{\text{sort}}, \cdots, \Lambda_{i+d}^{\text{sort}}\}$
  **end if**
  $i = i + 1$
**end while**
**Return:** $\Lambda_1$, $gap$

27. **IV. Extensions and future work.** As Rev. 2 points out, the spherical case can serve as a building block towards more
28. complex models. An ongoing work of the authors is the extension to the Euclidean unit ball where nodes closer to the
29. border will be more connected than the nodes closer to the center, allowing for more interesting applications. We agree
30. with Rev. 3 that graphex models will be worth exploring to extend our method to the sparse case.

31. **References**

32. [1] S. Bubeck, J. Ding, R. Eldan, and M. Rácz. Testing for high dimensional geometry in random graphs. *Random*
33. *Structures and Algorithms*, 49:503–532, 2016.


[Meta-Review · NeurIPS 2019]

The reviewers generally found the paper's main idea original and interesting, and the discourse easy to follow. The rebuttal satisfied most of the reviewers' questions, and they thought the proposed changes would make the paper acceptable at NeurIPS. A few remaining points that should be addressed in the final draft: - still unclear how limiting the assumption of spherical embedding is, and the provided reference in the rebuttal doesn't really provide much insight - it would be good to include more compelling experimental results (at a minimum, a small case study on a real world graph)